# An 11-mer Synthetic Peptide Suppressing Aggregation of Aβ25-35 and Resolving Its Aggregated Form Improves Test Performance in an Aβ25-35-Induced Alzheimer’s Mouse Model

**DOI:** 10.3390/biom14101234

**Published:** 2024-09-29

**Authors:** Rina Nakamura, Akira Matsuda, Youichirou Higashi, Yoshihiro Hayashi, Motomi Konishi, Motoaki Saito, Toshifumi Akizawa

**Affiliations:** 1Department of Pharmacology, Kochi Medical School, Kochi University, Kohasu, Oko-cho, Nankoku 783-8505, Kochi, Japan; r.nakamura@kochi-u.ac.jphigasi@kochi-u.ac.jp (Y.H.); saitomo@kochi-u.ac.jp (M.S.); 2O-Force Co., Ltd., 3454 Irino Kuroshio-cho, Hata-gun 789-1931, Kochi, Japan; jm-hayashiy@kochi-u.ac.jp; 3Laboratory of Medicinal and Biochemical Analysis, Faculty of Pharmaceutical Sciences, Hiroshima International University, 5-1-1, Hirokoshingai, Kure 737-0112, Hiroshima, Japan; matsuda@hirokoku-u.ac.jp; 4Equipment Support Planning Office, Kochi University, Kohasu, Oko-cho, Nankoku 783-8505, Kochi, Japan; 5Department of Integrative Pharmacy, Faculty of Pharmaceutical Sciences, Setsunan University, 45-1 Nagaotoge-cho, Hirakata 573-0101, Osaka, Japan; motomi@pharm.setsunan.ac.jp

**Keywords:** Alzheimer’s disease, amyloid-β peptide, YS-RD11 peptide, aggregation, ThT assay

## Abstract

There is a high demand for the development of drugs against Alzheimer’s disease (AD), which is related to the misfolding and aggregation of Amyloid-β (Aβ), due to the increasing number of patients with AD. In our present study, we aimed to assess the aggregation inhibitory effect of various synthetic YS-peptides on Aβ25-35 to identify an applicable peptide for clinical use for AD treatment and prevention. Suppression and aggregate resolution activities of YS-peptides against Aβ25-35 were evaluated using a Thioflavin T assay and scanning electron microscopy (SEM). Structure–activity relationship studies revealed that YS-RD11 (RETLVYLTHLD) and YS-RE16 (RETLVYLTHLDYDDTE) showed suppression and aggregate-resolution activities. The effect of YS-peptides on phagocytosis in microglial cells (BV-2 cells) demonstrated that YS-RD11 and YS-RE16 activated the phagocytic ability of microglia. In the Aβ25-35-induced AD mouse model, YS-RD11 prevented and improved the deficits in short-term memory. In conclusion, YS-RD11 is a suitable candidate therapeutic drug against AD and uses a strategy similar to that used for antibodies.

## 1. Introduction

Alzheimer’s disease (AD) is the most common age-related neurodegenerative disorder worldwide. Amyloid-β (Aβ) 42, which is produced by the cleavage of amyloid precursor protein (APP) by β- or γ-secretases, is considered to cause AD [1,2,3]. Since Aβ42 oligomers exhibit strong neurotoxicity, Aβ42 is a potential target for drug therapies against AD [4,5,6,7,8]. Hence, inhibitors against β- or γ-secretases and against Aβ42 oligomerization have been targeted to develop drugs against AD; however, there are no reports of their effectiveness on established cases of AD [9,10,11,12,13,14,15]. Therefore, several researchers have questioned the validity of the amyloid hypothesis. Currently, the primary focus is on the development of Aβ antibodies and vaccines aimed at inhibiting Aβ aggregation as potential therapeutic agents against AD. Studies on aducanumab proved that the amyloid hypothesis is one of the main causes of AD [6,16,17]. Lecanemab is the latest Aβ antibody drug approved by the U.S. Food and Drug Administration [17]. The mechanism of action of lecanemab involves the inhibition of Aβ aggregation, similar to that of aducanumab. This drug is potentially effective in the early stages of AD but is less effective in the later stages. In addition, the side effects of lecanemab, such as cerebral edema and cerebral microhemorrhage, are risk factors for its clinical use [8,18,19,20,21,22]. Thus, new, safe, and low-cost strategic drugs are required. Recent evidence suggests that in patients with AD, Aβ25-35 is produced via the enzymatic cleavage of Aβ42, and that Aβ25-35 can induce alterations in neuronal activity along with damage to long-term memory [23,24]. Furthermore, administration of Aβ25-35 in the CA1 subfield of the rat hippocampus induces morphological changes in the granular cells of the dentate gyrus as well as impairment in memory retrieval [25]. Several researchers have reported that Aβ25-35 undergoes a conformational change from a soluble to an aggregated β-structure form [26,27]. Thus, the study of the Aβ25-35 peptide has contributed considerably toward understanding the effects of Aβ toxicity and aggregation mechanisms.

We had previously reported that some peptides activate unfold-pro matrix metalloproteinase-7 (u-proMMP-7), after randomly screening nearly 1000 synthetic peptides in our peptide library [28]. One of these peptides, JAL-AK22 (KWEGHWYPEKPYKGSGFRCIHI), activates proMMP-7 by cleaving the pro-domain of MMP-7. Furthermore, the structure–activity relationship of JAL-AK22 proved that JAL-TA9 (YKGSGFRMI) showed the strongest auto-proteolytic activity and cleaved Aβ42 with serine protease-like activity. We use the term Catalytide (catalytic peptide) to denote proteolytic peptides [28]. Thus far, several peptides have been identified as Catalytides. Among them, JAL-TA9 and ANA-TA9 cleaved both the authentic soluble form Aβ42 (a-Aβ42) and the solid insoluble form Aβ42 (s-Aβ42) in the central region [29,30,31]. Examination of the route of administration revealed that ANA-TA9, one of the Catalytides, was more efficiently delivered to the brain by nasal application than by other routes [32].

After expanding the screening process using u-proMMP7 to 3000 types of peptides, we identified YS-RE20, which activates u-proMMP7 by acting as a chaperone rather than Catalytide. Chaperones are potential drug candidates for folding diseases, such as AD and Parkinson’s disease, which are caused by protein aggregation. In recent years, research has focused on low-molecular-weight chemical chaperones rather than on chaperone proteins [33,34,35,36]. Biochemical methods are used to screen for chemical chaperones with the aim of developing therapeutic agents for lysosomal diseases, which are folding diseases [33,37]. Screening of peptides with chaperone activity identified a peptide that suppresses the thermal aggregation of proteins [38,39].

In this study, Aβ25-35 was used to evaluate the aggregation inhibitory effect of a peptide created based on YS-RE20 (YS-peptides) because it is the shortest fragment peptide of APP that forms aggregates like Aβ42, exhibits neurotoxicity, and is known to show cognitive dysfunction when administered intracerebroventricularly [26,40] using Thioflavin-T (ThT) assay and scanning electron microscopy (SEM). We also evaluated the effect of YS-peptides against Aβ25-35-induced AD model mice with the aim of developing a therapeutic agent for AD.

## 2. Materials and Methods

### 2.1. Preparation of Peptides

YS-peptides and Aβ25-35 were synthesized from Fmoc-protected L-amino acid derivatives using an automated peptide synthesizer (433A; Applied Biosystems, Waltham, MA, USA; 0.1 mmol scale with a preloaded resin) and purified according to the method described by Nakamura et al. [28,31]. In brief, each peptide was purified using reverse-phase high-performance liquid chromatography (HPLC; Capcell Pak C18 column, SG, 10 mm i.d. × 250 mm; Shiseido Co., Ltd., Tokyo, Japan) with a linear gradient elution from 0.1% TFA to 50% CH_3_CN containing 0.1% TFA over 30 min. The purity of the synthetic peptides was confirmed using analytical reverse-phase HPLC (Capcell Pak C18 column, MGII, 4.6 mm i.d. × 150 mm; Shiseido Co., Ltd., Tokyo, Japan) monitored using a photodiode-array detector (SPD-M20A; SHIMADZU, Kyoto, Japan). Purified YS-peptides were characterized using ESI-MS using a Qstar Elite Hybrid LC-MS/MS system (Applied Biosystems, Framingham, MA, USA).

### 2.2. ThT Assay

To perform the ThT assay to evaluate aggregation, Aβ25-35 (final concentration of 100 μM) was incubated with ThT solution (final concentration of 10 μM) in Phosphate-buffered saline (PBS), pH 7.4. The ThT signal was monitored by measuring the fluorescence emission at 480 nm for 10 s when excited at 444 nm using Cytation 5 (BioTek, Vermont, WI, USA) at 37 °C [41]. Aggregated Aβ25-35 (100 μM) peptides were prepared by incubating for 18 h in PBS.

### 2.3. SEM

Aβ25-35 (100 μM) was incubated with YS-RD11 or YS-RE16 in PBS at 37 °C. The reaction mixture was centrifuged at 9000 rpm for 5 min. The supernatant was removed, 200 μL of MilliQ water was added and mixed, and the solution was centrifuged at 9000 rpm for 5 min to obtain a sample for SEM [42]. Aggregated Aβ25-35 (100 μM) peptides were prepared by incubating for 18 h in PBS.

### 2.4. Cell Culture and Viability

The human lung cancer cell line, A549, was obtained from the Riken Cell Bank (Ibaraki, Japan). The cells were cultured as described previously [30,43]. Briefly, A549 cells (4 × 10^3^ cells/well) were seeded into 96-well plates and maintained in Dulbecco’s modified Eagle’s medium (DMEM) supplemented with 10% fetal bovine serum, 100 units/mL penicillin, and 100 μg/mL streptomycin at 37 °C in a humidified atmosphere with 5% CO_2_. After incubation for 24 h, the medium was replaced and the cells were treated with 0.1 mg/mL YS-peptide, 0.1 mg/mL FLAG peptide, and 4 μM cisplatin (CDDP). After incubation for 72 h, the medium was exchanged for 110 μL of medium containing 10 μL WST-8 (water-soluble tetrazolium salt WST-8) reagent and 100 μL DMEM). The cells were further incubated for 1 h, the absorbance was determined at 450 nm with a reference wavelength of 620 nm using a SPECTRA max Plus384 microplate reader (Molecular Devices, LCC, San Jose, CA, USA).

### 2.5. Phagocytosis Assay

The microglia cell line, BV-2, was maintained in DMEM, supplemented with 5% fetal bovine serum in a CO_2_ incubator. BV-2 cells were harvested from the 24-well plates (4 × 10^4^ cells/well) and incubated for 48 h. The medium was then replaced with serum-free DMEM, and the cells were treated with 0.1 mg/mL YS-RD11 or YS-RE16. After incubation for 24 h, the cells were further incubated with 1 μm yellow-green carboxylate latex beads (Polysciences, Warrington, PA, USA; 1:5000 dilution) for 30 min in a CO_2_ incubator. Thereafter, the cells were fixed in 4% paraformaldehyde for 20 min. Bead uptake was evaluated in three randomly selected fields from four wells for each experiment using a laser confocal microscope [41,44].

### 2.6. Animals

All procedures met the guidelines of the U.K. Animals for Scientific Procedures and Directive 2010/63/EU of the European Parliament and the National Institutes of Health Guide for the Care and Use of Laboratory Animals and were approved by the Committee for the Care and Use of Laboratory Animals at Kochi University (permission number: L-00048). Twenty-eight male ICR mice, a strain of albino mice originating in Switzerland (4 weeks old; Japan SLC, Hamamatsu, Japan), were housed per cage and maintained at a controlled temperature (23 ± 1 °C) and humidity (55 ± 2%) and a constant day/night rhythm (14/10 h light/dark cycle; lights on at 05:00) with free access to water and food.

### 2.7. Intraventricular Injection of Aβ25-35 and YS-RD11

ICR mice were anesthetized with 1–3% isoflurane in a 75:25 mixture of nitrous oxide and oxygen. Mice were administered a stereotaxic injection of saline, 3 mM Aβ25-35, or a mixture of 3 mM Aβ25-35 and 3 mM YS-RD11 in the right lateral ventricle (anteroposterior, 0.2; mediolateral, 1.0; dorsoventral, 2.5 mm; from the bregma and cortical surface) using a 10 µL Hamilton syringe in a final volume of 3 μL per mouse, as previously described [41,45]. Aβ25-35 dissolved immediately before injection in order to inject in a nonaggregate form.

### 2.8. Intranasal Injection of YS-RD11

YS-RD11 diluted in PBS was administered to the left nostril of a male ICR mouse (*n* = 7) at a dose of 0.1 mg (10 μL of 10 mg/mL). Fourteen days after Aβ25-35 i.c.v. administration, YS-RD11 was administered once every three days for a total of four doses.

### 2.9. Spontaneous Alternation in Y-Maze Test

The behavior of mice in a spontaneous alternation Y-maze (40 cm long, with three arms positioned at equal angles) was observed to measure short-term spatial memory deficits. Mice were placed at the end of one arm and allowed to explore freely during a 10 min session, while the series of arm entries were recorded. An alternation was considered to occur if a mouse entered an arm that was distinct from the two previously entered arms. The percentage of relative alternations was calculated as follows: [number of alternations/(number of total arms entries 2)] × 100.

### 2.10. Statistical Analysis

All data are expressed as mean ± standard deviation of the mean. Differences were considered significant at *p* < 0.05. Significant difference tests were performed with Ordinary one-way ANOVA (Dunnett’s multiple comparisons test). Graphs were drawn and *p*-values were calculated using GraphPad Prism [version 9.5.1 (528)] (GraphPad Software, LLC., Boston, MA, USA).

## 3. Results

### 3.1. Screening for Effective Peptides

Five peptides of various lengths and regions were synthesized based on YS-YE20 (YKNMRETLVYLTHLDYDDTE) (Table 1). The aggregation inhibitory effect of these five peptides on Aβ25-35, an essential domain of Aβ42 aggregation, was first evaluated by fluorescence intensity using ThT.

In the reaction solution containing only Aβ25-35, high fluorescence intensity was observed at 4 and 24 h after the start of incubation, confirming that aggregation had occurred. Fluorescence intensity also increased in the reaction solution containing YS-YR5 (YKNMR) and YS-DE6 (DYDDTE). In contrast, the fluorescence intensity of solutions containing YS-RD11 (RETLVYLHTHL) and YS-RE16 (RETLVYLTHLDYDDTE) did not increase, indicating that they inhibited the aggregation of Aβ25-35 (Figure 1a). Moreover, YS-YD15, which contained YS-RD11, showed weak inhibitory activity. These results suggest that YS-RD11 is the smallest unit of the peptide that shows aggregation inhibitory activity against Aβ25-35 (Figure 1a and Table 1).

We also evaluated inhibitory effects in a time-dependent manner. The fluorescence intensity of the reaction mixture of peptide and Aβ25-35 was traced for 120 min at 15 min intervals. The fluorescence intensity of Aβ25-35 alone and reaction mixtures of Aβ25-35 and YS-YR5, YS-DE6, or YS-YD15 increased in a time-dependent manner (Figure 1b). Contrastingly, no increase in fluorescence intensity was observed in the presence of YS-RD11 and YS-RE16, indicating that these two peptides inhibited the formation of Aβ25-35 aggregates (Figure 1b). These results indicated that YS-RD11 and YS-RE16 are suitable seed peptides for the development of AD drugs.

### 3.2. Inhibitory Effect of YS-RD11 and YS-RE16 on Aβ 25-35 Aggregation

It was revealed that YS-RD11 and YS-RE16 inhibited the aggregation of Aβ25-35 at concentrations equivalent to those of Aβ25-35. Therefore, we evaluated the concentration-dependent aggregation inhibitory effect of YS-RD11 and YS-RE16 on Aβ25-35. Various concentrations of the peptides (YS-RD11; 1 to 100 µM, YS-RE16; 0.01 to 10 µM) were incubated independently with 100 µM Aβ25-35 (final con. 100 µM) in PBS at 37 °C. The fluorescence intensity of reaction mixtures was measured after 4 h. Compared to Aβ25-35 alone, a concentration-dependent decrease in fluorescence intensity of YS-RD11 was observed, with a significant decrease in fluorescence intensity at concentrations above 5 µM (Figure 2a). Furthermore, the fluorescence intensity of YS-RE16 significantly decreased at concentrations above 0.5 µM (Figure 2b). The IC_50_ values of YS-RD11 and YS-RE16 were ~1.5 µM and ~0.5 µM, respectively.

We then investigated the effect of YS-RD 11 and YS-RE16 on the amyloid formation of Aβ25-35 by observing its shape using SEM. Aβ25-35 alone and YS-RD11 or YS-RE16 were reacted with Aβ25-35 at 37 °C in PBS for 4 h. The reaction mixture was photographed using SEM. In the case of Aβ25-35 alone, amyloid fibrils were observed (Figure 3a). In contrast, the number of amyloid fibrils decreased in the presence of YS-RD11, and almost no amyloid fibrils were observed in the presence of YS-RE16 (Figure 3b,c). These results indicated that the inhibitory effect against fibrillization of Aβ25-35 of YS-RE16 was more potent than that of YS-RD11. Furthermore, quantification of amyloid fibrils revealed that both YS-RD11 and YS-RE16 significantly reduced the number of fibrils compared to Aβ25-35 alone (Figure 3d). The IC_50_ value of YS-RE16 was one-tenth that of YS-RD11 and fibril formation was almost completely suppressed. These results demonstrated that YS-RE16 has a stronger inhibitory effect on Aβ25-35 aggregation than YS-RD11.

### 3.3. Resolving Effects on Aggregated Aβ 25-35

Next, we evaluated the resolution activities of YS-RD11 and YS-RE16 against aggregated Aβ25-35. First, Aβ25-35 was incubated for 18 h to initiate the formation of an aggregate of Aβ25-35 (AgAβ25-35). Then, various concentrations (1 to 100 µM) of YS-RD11 and YS-RE16 were added and co-incubated with AgAβ25-35 for 5 days. The fluorescence intensity was measured to evaluate amyloid fibril resolution activity. YS-RD11 showed a significant decrease in fluorescence intensity at 5 µM and 10 µM compared to Aβ 25-35 alone (Figure 4a). In contrast, YS-RE16 showed a concentration-dependent decrease in fluorescence intensity, with a significant decrease at 100 µM (Figure 4b). The resolving potency of YS-RD11 in 5 µM was higher than that of YS-RE16.

Next, the resolving effect of YS-RD11, YS-RE16, and YS-DE6 on aggregated Aβ25-35 was assessed by observing its shape using SEM. YS-DE6, as the negative control, did not show any effect against Aβ25-35 aggregation. Both Aβ25-35 alone and co-incubation with YS-DE6 demonstrated a cluster of amyloid fibrils (Figure 5a,b). In contrast, incubation with 5 µM YS-RD11 showed a significant reduction in the number of amyloid fibrils compared to those of Aβ25-35 alone (Figure 5c). In the presence of YS-RE16, the number of amyloid fibrils decreased slightly (Figure 5d). The quantification of amyloid fibrils revealed that their number was significantly reduced in the presence of YS-RD11 compared to that in Aβ25-35 alone (Figure 5e). These data indicated that the resolving activity against fibrillization of Aβ25-35 of YS-RD11 was more potent than that of YS-RE16.

### 3.4. Cell Experiments

Cytotoxicity is a critical adverse effect associated with clinical use. Thus, we examined the effects of YS-RD11 and YS-RE16 on the growth of A549 cells prior to the in vivo experiments (Figure 6a). YS-RD11 and YS-RE16 did not show a significant inhibitory effect on the growth of A549 cells compared with the FLAG peptide, which was used as a peptide control. In contrast, the chemotherapeutic agent, CDDP, which was used as the positive control, inhibited the growth of A549 cells. These results indicated that the YS-peptides did not show cytotoxicity. Interestingly, both YS-RD11 and YS-RE16 promote the cell proliferation of A549 cells.

We analyzed the effect of the YS-peptides on the phagocytic activity of BV-2 cells. BV-2 cells were treated with 0.1 mg/mL YS-peptide prior to the administration of yellow-green carboxylate beads. Bead uptake in BV-2 cells treated with YS-RD11 and YS-RE16 was significantly higher than that in non-treated cells (Figure 6b). These results revealed that YS-RE16 activated the phagocytic ability of microglia to remove waste products from the brain.

### 3.5. Animal Experiments

Based on the results obtained from the in vitro experiments, YS-RD11 and YS-RE16 were considered suitable AD drug candidates. YS-RD11 was considered a potential therapeutic agent for AD because of its high solubility and ability to disperse aggregates at low concentrations. Thus, we investigated whether YS-RD11 improved the short-term spatial memory deficits induced by Aβ25-35 i.c.v. administration. As previously reported, Aβ25-35 i.c.v. administration AD model mice have shown that aggregated Aβ25-35 and administration of non-aggregated Aβ25-35 dissolved immediately before use resulted in cognitive dysfunction within one and two weeks, respectively [41,46].

First, we evaluated whether YS-RD11 prevents the short-term spatial memory deficits induced by Aβ25-35 i.c.v. injection. After Aβ25-35 injection or co-injection of YS-RD11 and Aβ25-35, the spontaneous alternation rate was measured using the Y-maze test (Figure 7a). On day 14, the Aβ25-35-co-injected YS-RD11 group showed a significantly heightened alternation rate compared with that of the Aβ25-35 group (Figure 7b). However, on day 28, the alternation ratio of YS-RD11 co-injected with the Aβ25-35 group decreased to that observed in the Aβ25-35 group (Figure 7c). These results indicate that the inhibitory effect of Aβ25-35 on cognitive decline is observed at 14 days but disappears thereafter.

Therefore, we evaluated its therapeutic effect using the Y-maze test by administering YS-RD11 intranasally (i.n.), which is a less invasive route of administration and can be administered multiple times. YS-RD11 was administered once every three days for a total of four doses starting on day 14 when cognitive deficit induced by Aβ25-35 i.c.v. administration was confirmed (Figure 7a). On day 28, the alternation rate of the i.n. injection group was similar to that of the saline group and significantly higher than that of the Aβ25-35 group (Figure 7c). These results indicated that YS-RD11 improved the deficits in short-term memory induced by Aβ25-35.

## 4. Discussion

Although many trials have been conducted to develop AD treatment drugs, including natural products and peptides, the results have not been encouraging [15,47,48]. Thus, the current main strategy is to develop Aβ antibodies and vaccines aimed at inhibiting Aβ aggregation. Recently, aducanumab, an antibody drug that inhibits Aβ aggregation, has been approved by the U.S. Food and Drug Administration. Aducanumab has been shown to inhibit the clinical manifestation of mild AD; however, drugs that radically eliminate Aβ and improve AD symptoms are severely lacking. Lecanemab, the latest Aβ antibody, is expected to be more effective than aducanumab with a similar mechanism [17]. Although lecanemab is potentially effective in the early stages of AD, its effect in later stages is doubtful. In addition, the development costs and side effects of antibody drugs such as lecanemab pose problems.

In the present study, we assigned the usefulness of a small peptide, and identified two kinds of peptides, YS-RD11 and YS-RE16, which showed inhibitory activity against the aggregation of Aβ25-35 and resolving activity against AgAβ25-35. The IC_50_ values indicated that YS-RE16 possessed higher inhibitory activity against Aβ25-35 aggregation than YS-RD11 (Figure 3). However, on the promotion of resolving activity against aggregated Aβ25-35, the potency of YS-RD11 was 10 times higher than that of YS-RE16 (Figure 6). These results suggest that their resolving mechanisms differ.

SEM analysis proved that YS-RD11 and YS-RE16 not only inhibited further fibril formation from protofibrils but also promoted the resolution of fibrils (Figure 4 and Figure 7). These data strongly support the results obtained from the ThT assay and suggest that YS-RD11 and YS-RE16 may be effective in improving AD at both early and late stages. This is one of the excellent features of YS-RD11 and YS-RE16 compared with lecanemab.

In the cell experiment, YS-peptides did not show cytotoxicity on the growth of A549 cells (Figure 6a), suggesting that, similar to JAL-TA9 and ANA-TA9, YS-RD11 and YS-RE16 are also safe for patients. The remaining question is the mechanism by which the soluble Aβ, as a result of treatment with YS-RD11 and YS-RE16, is removed from the brain. Recently, the relationship between microglial phagocytosis and age-related neurodegenerative disorders has been investigated. Activated microglial phagocytosis can decrease AD progression; thus, microglia are a potential therapeutic target for AD treatment [49,50,51,52]. The current study revealed that YS-RD11 and YS-RE16 activated the phagocytic ability of BV-2 cells (Figure 6b), suggesting that both peptides possess the ability to remove solubilized Aβ by promoting phagocytosis of microglia.

In the animal experiments, i.c.v. co-injection of Aβ25-35 and YS-RD11 suppressed the cognitive decline induced by i.c.v. injection of Aβ25-35. However, the clinical application of i.c.v. administration is difficult because of its highly invasive nature, with the aim of developing therapeutic agents for AD. When targeting AD, drug delivery to the brain is essential; however, the presence of the BBB limits their entry into the brain from the systemic circulation, which poses a problem for drug delivery by i.v. administration. To address this, studies have established a direct and efficient drug delivery of the peptide oxytocin and CPN-116 to the brain via nasal drug application [53]. Therefore, a direct route from the nasal cavity to the brain has been gaining increasing attention.

Recently, we reported that the brain concentration of ANA-TA9, which consists of nine amino acids, was higher after nasal administration than after intraperitoneal administration, despite a much lower plasma concentration after nasal administration. Similar findings were observed for transport to the CSF after nasal and intravenous administration. These results demonstrate the usefulness of i.n. administration as a route for the administration of small-molecule peptides targeting the brain [32].

Therefore, YS-RD11, which was administrated i.n. to mice with impaired cognitive function induced by an i.c.v. injection of Aβ25-35, resulted in improved Y-maze test performance. In this study, we did not quantitatively analyze YS-RD11 in the brain after nasal administration. However, these results suggested that YS-RD11, similar to ANA-TA9, was effective in the brain after intranasal administration.

## 5. Conclusions

In this study, we proved that YS-RD11 and YS-RE16 inhibited Aβ25-35 aggregation and resolved AgAβ25-35. In addition, YS-RD11 co-injection with Aβ25-35 i.c.v. suppressed the short-term spatial memory deficits induced by Aβ25-35. Furthermore, YS-RD11 i.n. administration improved the short-term spatial memory deficits induced by Aβ25-35 in AD model mice. This study led to the hypothesis that YS-RD11 prevents memory deficit by suppressing aggregation of Aβ25-35.

These results indicate that YS-RD11 is effective against both the early and later stages of AD (Figure 8).

Further studies evaluating the inhibitory effects of YS-RD11 on Aβ42 or Aβ40 aggregation and improvement in short-term spatial memory deficits using APP-knock-in model mice are warranted. Nevertheless, YS-RD11 and YS-RE16 can overcome problems associated with antibody drugs, such as manufacturing costs and side effects, and may be applicable as AD drugs with a less invasive and more versatile route of administration. We believe that YS-RD11 may be an attractive drug candidate manufactured with low costs for clinical use via i.n. injection.

## 6. Patents

T. Akizawa, M. Saito, and R. Nakamura, 2022, AMYLOID-β AGGREGATION INHIBITOR, PHARMACEUTICAL COMPOSITION FOR AMYLOID-β AGGREGATION DISEASES, AND USE APPLICATION OF SAME, WO/2022/239765.

## Figures and Tables

**Figure 1 biomolecules-14-01234-f001:**
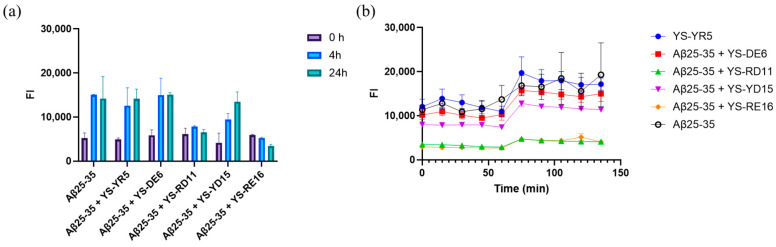
ThT fluorescence profile of reaction mixtures with Aβ25-35 and YS- peptides: (**a**) Five kinds of 100 μM YS-peptides (YS-YR5: YKNMR, YS-DE6: DYDDTE, YS-RD11: RETLVYLTHLD, YS-YD15: YKNMRETLVYLTHLD, and YS-RE16: RETLVYLTHLDYDDTE) were incubated in PBS. Fluorescence intensity was measured at 0, 4, and 24 h. (**b**) Time-course aggregation of Aβ25-35; 100 μM Aβ25-35 was incubated in PBS and fluorescence intensity was measured for 120 min at 15 min intervals. Data shown are mean ± standard deviation.

**Figure 2 biomolecules-14-01234-f002:**
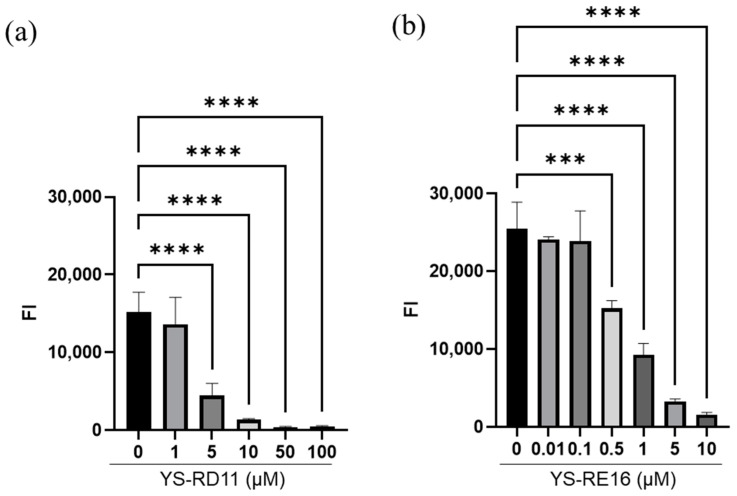
Effect of YS-RD11 and YS-RE16 on Aβ25-35 aggregation. Various concentrations of (**a**) YS-RD11 (1, 5, 10, 50, and 100 µM) or (**b**) YS-RE16 (0.01, 0.1, 0.5, 1, 5, and 10 µM) were co-incubated with 100 µM Aβ25-35 in PBS at 37 °C for 4 h. Data shown are mean ± standard deviation (*n* = 3), versus Aβ group. *** *p* < 0.001, **** *p* < 0.0001. 0.5 µM YS-RE16, *p* = 0.0002.

**Figure 3 biomolecules-14-01234-f003:**
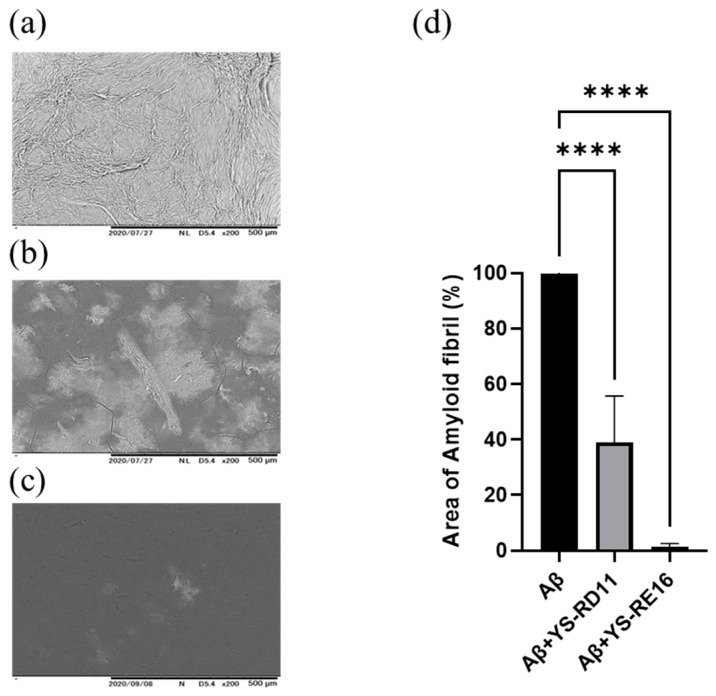
Aβ25-35 aggregate reduction effect of YS-RD11 and YS-RE16 using SEM: (**a**) 100 µM Aβ25-35, (**b**) 100 µM Aβ25-35 co-incubated with 100 µM YS-RD11, (**c**) 100 µM Aβ25-35 co-incubated with 100 µM YS-RE16, (**d**) quantification of amyloid fibrils. Data shown are mean ± standard deviation (*n* = 6), versus Aβ group. **** *p* < 0.0001.

**Figure 4 biomolecules-14-01234-f004:**
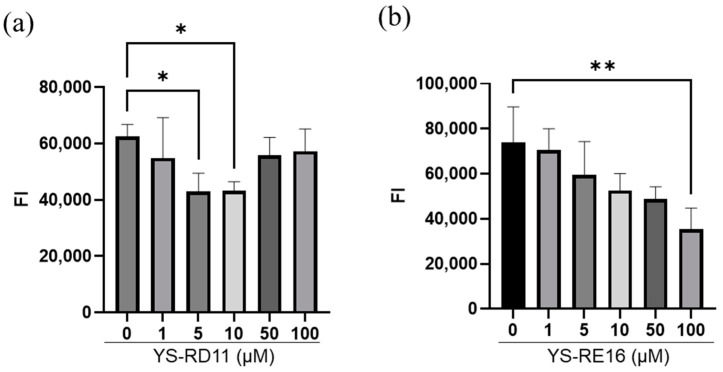
Concentration dependency of YS-RD11 and YS-RE16 against aggregated Aβ25-35. Various concentrations of (**a**) YS-RD11 or (**b**) YS-RE16 (1, 5, 10, 50, and 100 µM) were co-incubated with 100 µM Aβ25-35 in PBS at 37 °C. The fluorescence intensity was measured after co-incubation for 5 days. Data shown are mean ± standard deviation (*n* = 3), versus Aβ group. * *p* < 0.05, ** *p* < 0.01. 5 µM YS-RD11, *p* = 0.041; 10 µM YS-RD11, *p* = 0.045; 100 µM YS-RE16, *p* = 0.005.

**Figure 5 biomolecules-14-01234-f005:**
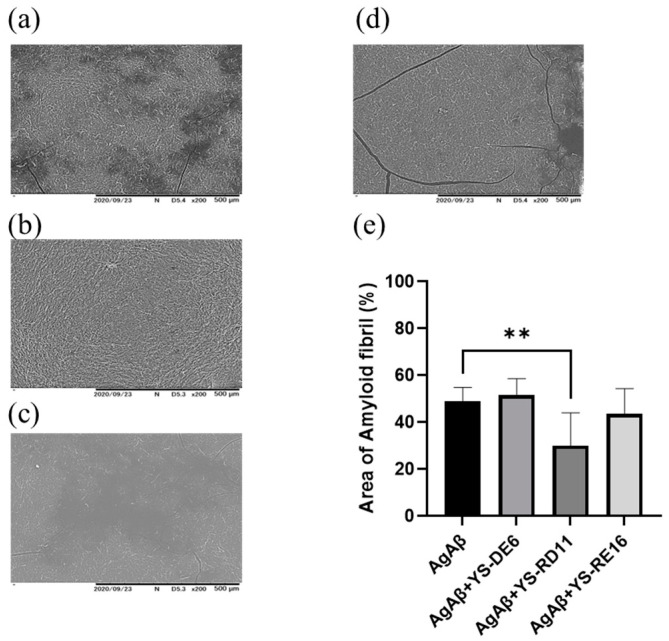
Assessment of the AgAβ25-35-resolving effect of YS-RD11 and YS-RE16 using SEM: (**a**) 100 µM Aβ25-35, (**b**) 100 µM Aβ25-35 co-incubated with 100 µM YS-DE6, (**c**) 100 µM Aβ25-35 co-incubated with 5 µM YS-RD11, (**d**) 100 µM Aβ25-35 co-incubated with 100 µM YS-RE16. (**e**) Quantification of amyloid fibrils. Data shown are mean ± standard deviation (*n* = 6), versus Aβ group. ** *p* < 0.01. AgAβ + YS-RD11, *p* = 0.009.

**Figure 6 biomolecules-14-01234-f006:**
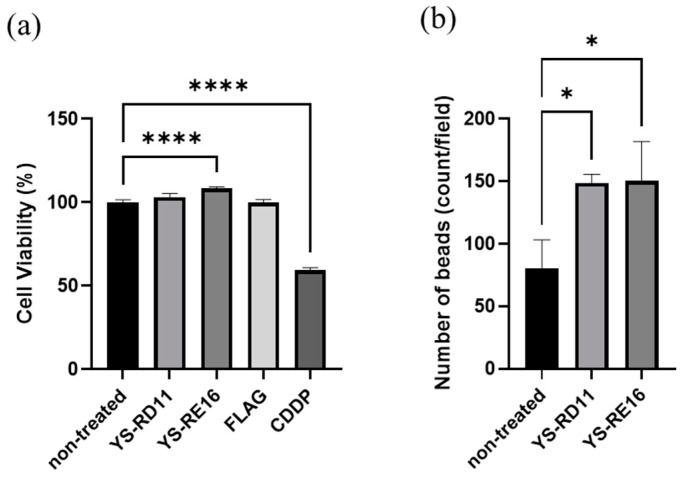
Effect of YS-RD11 and YS-RE16 on the cells: (**a**) The relative viability of A549 cells cultured in the presence or absence of 0.1 mg/mL peptides (YS-RD11, YS-RE16, and FLAG peptide) for 72 h. CDDP (4 μM), an anticancer drug, was used as positive control. Cell viability was determined by WST-8 assay. Data shown are mean ± standard deviation (*n* = 4), versus non-treated group. **** *p* < 0.0001 (**b**) BV-2 cells were treated with PBS or 0.1 mg/mL YS-RD11 and YS-RE16 for 24 h. Data shown are mean ± standard deviation (*n* = 3), versus Aβ group. * *p* < 0.05. YS-RD11, *p* = 0.018; YS-RE16, *p* = 0.016.

**Figure 7 biomolecules-14-01234-f007:**
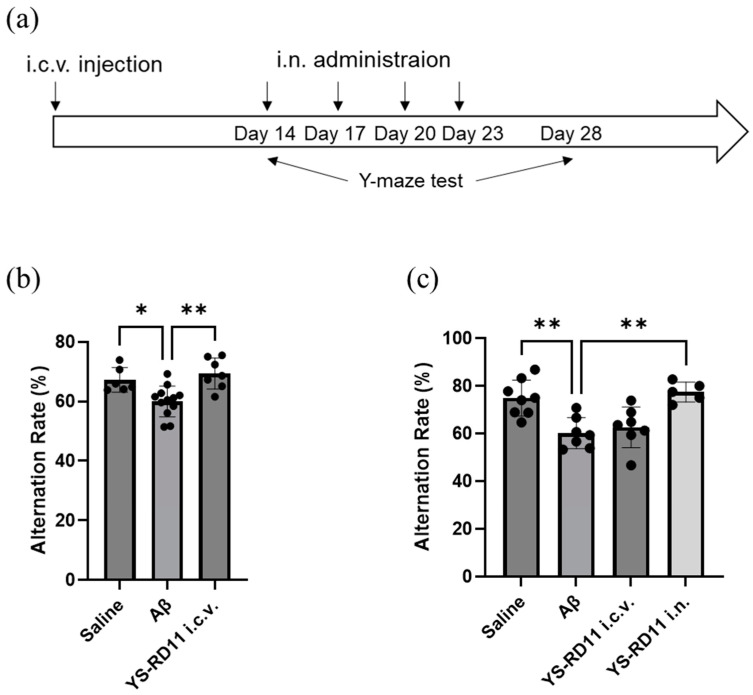
Effect of YS-RD11 on cognitive deficit in Aβ25-35 i.c.v.-induced AD model mice: (**a**) Experimental schedule. Y-maze test was performed at (**b**) 14 and (**c**) 28 days after i.c.v. injection. In the case of i.c.v. administration, saline, 9 nmol Aβ25-35, or a reaction mixture of 9 nmol Aβ25-35 and 9 nmol YS-RD11 was injected and 100 µg of YS-RD11 was injected i.n. Data shown are mean ± standard deviation (*n* = 7), versus Aβ group. * *p* < 0.05, ** *p* < 0.01. (**a**) Saline, *p* = 0.016; YS-RD11, *p* = 0.001; (**b**) Saline, *p* = 0.002, YS-RD11 i.n., *p* = 0.001.

**Figure 8 biomolecules-14-01234-f008:**
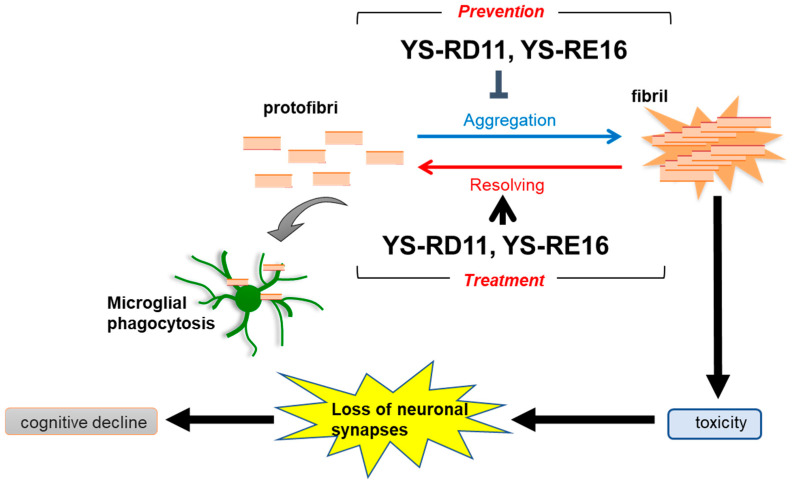
Effect of YS-RD11 and YS-RE16 on Aβ25-35 protofibril and fibril formation.

**Table 1 biomolecules-14-01234-t001:** Amino acids sequence of YS-YE20 and five kinds of peptides derived from YS-RE20.

	Sequence
YS-YE20	YKNMRETLVYLTHLDYDDTE
YS-YR5	YKNMR
YS-DE6	DYDDTE
YS-RD11	RETLVYLTHLD
YS-YD15	YKNMRETLVYLTHLD
YS-RE16	RETLVYLTHLDYDDTE

## Data Availability

The data and materials used in this study are available from the corresponding authors upon reasonable request.

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
