# Peer review of "An 11-mer Synthetic Peptide Suppressing Aggregation of Aβ25-35 and Resolving Its Aggregated Form Improves Test Performance in an Aβ25-35-Induced Alzheimer’s Mouse Model"

_biomolecules, 2024, doi:10.3390/biom14101234_

Round 1
Reviewer 1 Report
Comments and Suggestions for Authors
This study of a new peptide drug candidate for Alzheimer's disease (AD) has some merit but also has important weaknesses that are not discussed in the paper. The conclusions are overstated and not rigorously supported by the experimental design and results.
Major weaknesses/limitations:
1- Why was Abeta 25-35 used for the in vitro experiments rather than Abeta 1-42, which is the one being elevated in AD? Please justify this choice.
2- The mouse model used is not sufficiently described. What are ICR mice? Do you inject Abeta 25-35 in an aggregated form or monomeric form? Why did you inject this particular amount of Abeta 25-35? Do you know if Abeta 25-35 form aggregates in the brain? Does it diffuse out of the lateral ventricle? Was this model previously characterized in a publication? What is the time-course of memory deficit? How do you know that it is resulting from Abeta 25-35 aggregation?
3- The peptide drug YS-RD11 is not measured in the brain tissue after intranasal administration. Using an analytical method to detect and quantify the drug in the brain and matching it to measured outcome would have increased the scientific rigor of this study. A previous study with a different peptide is cited to support the hypothesis that YS-RD11 gets into the brain, but the absence of direct measurement of YS-RD11 in the brain is a major weakness that needs to be discussed in the paper.
4- There is a disconnect between the in vitro experiments showing anti-aggregation activity and pro-phagocytosis activity of the peptide drug(s) and the outcomes measured in vivo. Only behavioral outcomes are measured in vivo. In the absence of any biochemical measurements (amount of Abeta 25-35, amount of aggregation, activity of microglia), it is not possible to draw any conclusion about the mechanism of action of YS-RD11 in vivo, in the brain. Please clearly state that this study led to the hypothesis that this drug prevents memory deficit by suppressing aggregation.
Minor weaknesses:
1- Please indicate individual results by adding dots on your bar graphs, especially for in vivo results.
2- More details are needed for the experimental methods. For example, at what temperature did you run your ThT assay? For SEM, how long did you incubate Abeta 25-35 to form aggregates? What is the WST-8 reagent? What volume of Abeta 25-35 preparation did you inject icv? Please be consistent when indicating concentrations, rather than giving them by molarity sometimes and by mass/volume at other times.
3- Line 166 says YS-RE20 but in Figure 1, the Table indicates YS-YE20, please correct.
Reviewer 2 Report
Comments and Suggestions for AuthorsÂ
Â
